# Acute Esophageal Necrosis in Acute Pancreatitis—Report of a Case and Endoscopic and Clinical Perspective

**DOI:** 10.3390/diagnostics13030562

**Published:** 2023-02-03

**Authors:** Monica Grigore, Iulia Enache, Mirela Chirvase, Andrada Loredana Popescu, Florentina Ionita-Radu, Mariana Jinga, Sandica Bucurica

**Affiliations:** 1Department of Gastroenterology, “Carol Davila” University Central Emergency Military Hospital, 010825 Bucharest, Romania; 2Department of Gastroenterology, “Carol Davila” University of Medicine and Pharmacy, 020021 Bucharest, Romania

**Keywords:** acute esophageal necrosis, black esophagus, endoscopy, Gurvits syndrome

## Abstract

Esophageal stroke, also known as acute esophageal necrosis or Gurvits syndrome, is an entity that has gained more and more recognition in the last two decades. It is also named “black esophagus” because of striking black discoloration of the esophageal mucosa, with an abrupt transition to normal mucosa at the gastroesophageal junction. Its most common clinical presentation is represented by upper gastrointestinal bleeding and esophagogastroduodenoscopy is the main diagnostic tool. Among the etiopathogenetic and multiple predisposing factors described are hypovolemia, shock state, ischemia, congestive heart failure, acute renal failure, infections, trauma, and diabetes mellitus. Current management of this condition consists of treating the underlying pathology, nil per os, and antacid administration in uncomplicated cases. Although most of the cases have favorable prognosis, complications such as pneumomediastinum or esophageal stricture may occur and fatal cases are a consequence of underlying comorbidities.

**Figure 1 diagnostics-13-00562-f001:**
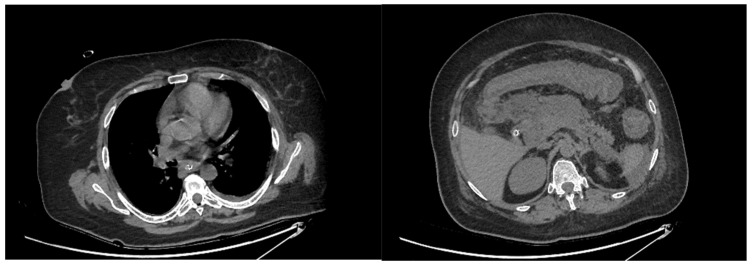
CT revealed minimal pleural effusion and confirmed the presence of peripancreatic necrosis and moderate ascites.

Once considered a rare entity, acute esophageal necrosis gained more awareness among endoscopists and became a diagnosis that should be considered in a certain group of patients with upper gastrointestinal bleeding. The reported prevalence in the literature of this syndrome varies between 0.06 and 0.28%. This syndrome is found predominately among men and the mean age at diagnosis varies between 67 and 76 years old between studies [1,2,3,4]. A 56-year-old female patient presented at the emergency room for intense abdominal pain, acute diarrhea, and emesis. Two weeks before, the patient was hospitalized in another medical center where she was treated for biliary acute pancreatitis and Cl. difficile infection. The physical exam was notable for alteration of her general status with pallor of the skin, oliguria, and hemodynamic instability, which required vasopressor support. The laboratory tests were remarkable for leukocytosis of 48,000/mm^3^, CRP of 250.45 mg/L, mild normochromic normocytic anemia with Hb of 9.1 g/dL, and positive toxin A/B for Cl. difficile. Computed Tomography (CT) showed the presence of necrosis around the pancreatic head and moderate ascites and minimal pleural effusion (Figure 1). Treatment with Vancomycin and Metronidazole was started with fluid resuscitation, taking into account the previous Cl. Difficile infection. Later on, the urinary culture that was extracted on the admission day proved to be positive for Pseudomonas aeruginosa and treatment with Colistin was added, according to antibiogram.

**Figure 2 diagnostics-13-00562-f002:**
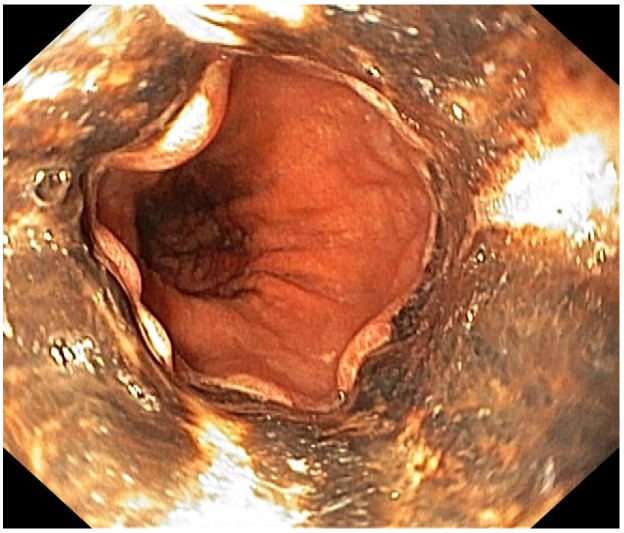
Black appearing mucosa of the esophagus that abruptly stops at the gastroesophageal junction. Five days later, the patient developed coffee ground emesis and the hemoglobin dropped from 9.3 g/dL to 8.8 g/dL. Upper endoscopy revealed a striking black appearance of two distal thirds of esophagus that abruptly stopped at the gastroesophageal junction, without other modifications (Figure 1 and Figure 2). Treatment with PPI, Sucralfate, and parenteral nutrition were initiated while she was kept nil per os.

**Figure 3 diagnostics-13-00562-f003:**
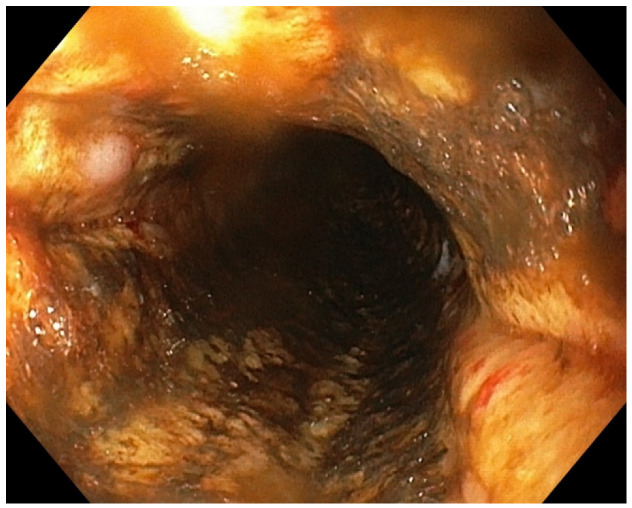
Circumferential blackening of the esophageal mucosa.

**Figure 4 diagnostics-13-00562-f004:**
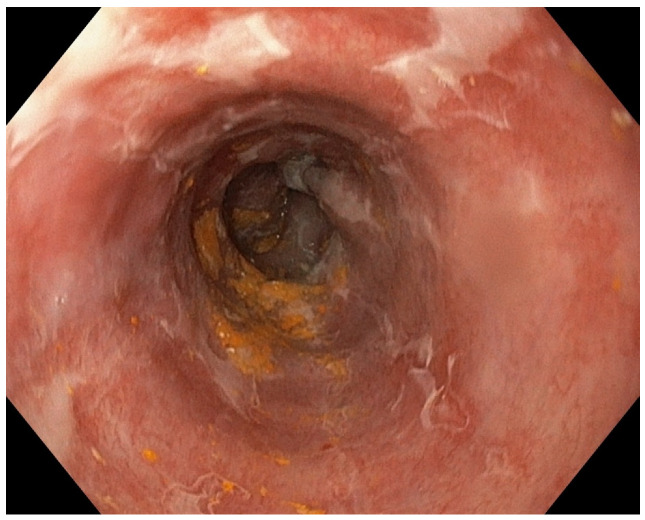
Endoscopic check-up revealed persistent ulcerations covered by white exudates that raised the suspicion of esophageal candidiasis. One week later, endoscopic check-up showed persistent esophageal ulcers covered by white exudates and biopsies were performed to exclude an esophageal candidiasis, but they were negative (Figure 4 and Figure 5).

**Figure 5 diagnostics-13-00562-f005:**
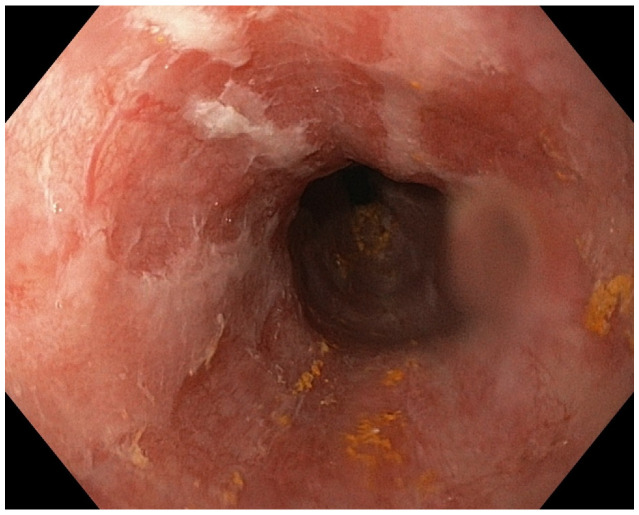
Second look-up showed white exudates covering the esophageal mucosa, just one week after the first endoscopy.

The evolution was slowly resolving: the initial hypovolemia was corrected with vasopressor support and vigorous fluid resuscitation. Given the Cl. difficile history and previous antibiotic therapy for biliary affliction, the infectious disease specialist recommended initial antibiotics that later were adjusted for Pseudomonas aeruginosa urinary infection sensitive to Colistin. The pancreatitis had a favorable evolution and the patient was treated conservative with PPI and parenteral nutrition until after the endoscopic second look. The oral nutrition was initiated with liquids, the patient condition improved significantly, and the patient was discharged. The patient did not present for follow-up. The clinical presentation of patients with black esophagus is in most cases represented by upper gastrointestinal bleeding related symptoms (melena, coffee-ground emesis, and hematemesis), but dysphagia, chest pain, epigastric pain, and vomiting can be experienced [1,4,5]. The etiology is multifactorial and is considered that acute esophageal necrosis generally results from a combination of tissue hypoperfusion, impaired defense mechanism and backflow injury from gastric acid. Several medical conditions such as sepsis, diabetes mellitus, hypertension, dyslipidemia, malignancy, and malnutrition were recognized as risk factors [6,7]. In our case, background pancreatitis complicated with Clostridioides infection and sepsis and additional emesis associated with hypovolemia (respective hypoperfusion) are possible risk factors for the emergence of “black esophagus” or acute esophageal necrosis.

Endoscopy is the main diagnostic tool, but computed tomography could be helpful if complications such as esophageal perforation, mediastinitis, or pneumomediastinum are suspected [8,9]. The distal third of the esophagus is the most commonly affected due to poor vascularization, but the rest of the esophagus can be affected to various degrees, depending on the severity of the initial injury. One possible explanation for the fact that the distal segment is more often affected resides in the diverse blood supply of the esophagus. The upper part is vascularized by descending branches of the inferior thyroid artery, the mid-esophagus vascularization comes from branches from the thoracic aorta, and the distal segment is served by branches from left gastric artery and left inferior phrenic artery [7]. The isolated involvement of the mid-esophagus is rare and was described in case reports of patients suffering from diseases of the thoracic aorta such as aortic dissection or aortic aneurysm [10,11]. It is thought that the prevalence of this entity is underestimated because of several reasons. The first reason is represented by the fact that even a temporary reduction in esophageal perfusion can determine extensive necrosis. Transitory hypotension that lasts about 30 min was described to cause ischemic insult of the esophagus in a case report. The complete resolution of the necrosis was described as early as 7 days [2,12]. Another reason is that timing of the index endoscopy can influence the reported lesions of the mucosa because there are several stages of mucosal healing that can mislead the endoscopist. This happened also in the case we presented, where an endoscopic check-up revealed several white exudates that were misinterpreted as esophageal candidiasis. In order to classify the natural progression of the disease, in 2007, Gurvits et al. proposed a staging system according to the macroscopic appearance of the esophageal mucosa. According to the staging system he proposed, stage 0 describes a pre-necrotic normal appearing esophagus, while stage 1 is represented by the classic aspect of black-appearing circumferential esophageal mucosa that ends abruptly at the gastroesophageal junction. Stage 2 is characterized by white exudates that are easily stripped off, revealing a pink friable mucosa, and stage 3 is represented by esophageal mucosa that recover its normal pink appearance [13]. The optimal timing for performing endoscopic control is yet to be decided, but the average period described until the resolution of necrosis was about from one to two and a half weeks [2,12]. The overall mortality reported in the literature in patients with black esophagus was 31.8%, but the mortality related to the esophageal disease was about 5.7% [12]. Regarding our case, we found a small number of similar cases with black esophagus associated with pancreatitis in the literature. In a series of 29 patients, only one presented esophageal stroke and pancreatitis [2], another case was reported associated with post-ERCP pancreatitis [14] and Gurvits reported only 4 cases of associated pancreatitis from 88 patients with esophageal acute necrosis found in a retrospective study between 1965 and 2006 [13]. Despite the impressive endoscopic appearance, most of the time the prognosis is good and there is a spontaneous resolution of the lesions. Generally, it is managed by conservative measures and the focus should be on the concomitant management of the concomitant conditions.

## Data Availability

Not applicable.

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
