# Peer review of "Acute Esophageal Necrosis in Acute Pancreatitis—Report of a Case and Endoscopic and Clinical Perspective"

_diagnostics, 2023, doi:10.3390/diagnostics13030562_

Round 1

Reviewer 1 Report

Interesting case of Gurvits syndrome with good endoscopic images.  You should include a hypothesis as to why your patient developed black esophagus and indicate if a clinical follow up was performed including a repeat endoscopy.  In the abstract, management with correction of underling medical conditions and antiacids/nil-per-os is standard, except for cases of perforation where surgical consult should be sought.  The title should include "Gurvits syndrome" as a common name of this condition.  

Author Response

Dear Reviewer,

Thank you for giving us the opportunity to improve our  manuscript “Acute Esophageal Necrosis (Gurvits Syndrome) - Endoscopic and Clinical Perspective” for publication in Diagnostics  special issue- Diagnosis and Prognosis of Gastrointestinal Diseases section Interesting images.

We appreciate the time and effort that you dedicated to provide feedback on our manuscript and we are grateful for the insightful comments on and valuable improvements to our paper and for considering interesting case of Gurvits syndrome with good endoscopic images.

We have incorporated your highly valuable suggestions in the manuscript, highlighted with track changes:

  1. You should include a hypothesis as to why your patient developed black esophagus and indicate if a clinical follow up was performed including a repeat endoscopy.

Response: Since the patients has a complex and severe pathology and the literature shows that are numerous factors implied in etiology of acute esophageal necrosis we think that are multiple factors that could led to development of acute esophageal necrosis in this case and we added in the manuscript:

“In our case the background pancreatitis complicated with Clostridioides infection and sepsis and additional emesis associated with hypovolemia ( respective hypoperfusion)are the possible risk factors for apparition of “black esophagus” or acute esophageal necrosis” . We included the follow-up endoscopy “One week later, endoscopic check-up showed persistent esophageal ulcers covered by white exudates” and included images. Regarding the follow-up the patient did not return and we added in the text: “The patient did not return for follow-up.”

  1. In the abstract, management with correction of underling medical conditions and antiacids/nil-per-os is standard, except for cases of perforation where surgical consult should be sought.

Response: According to specific suggestion we modified  in the  abstract “Current management of this condition consists of treating the underlying pathology, nil per os and antiacids administration in uncomplicated cases.”

  1. The title should include "Gurvits syndrome" as a common name of this condition. 

Response: According to valuable suggestion we added the common name of the condition in the title “Acute Esophageal Necrosis (Gurvits Syndrome) - Endoscopic and Clinical Perspective”

Sincerely yours, 

Sandica Bucurica

Reviewer 2 Report

ACUTE ESOPHAGEAL NECROSIS

The case / images presented are of interest but could be improved in the following aspects:

1.      The structure is difficult to read because there are no breaks throughout the text.

2.      It is a type of article whose importance corresponds to the image. There is a lot of text that should be removed and summarized for easier reading .

3.      I suggest providing a CT image

Author Response

Dear Reviewer,

Thank you for giving us the opportunity to improve our  manuscript “Acute Esophageal Necrosis (Gurvits Syndrome) - Endoscopic and Clinical Perspective” for publication in Diagnostics  special issue- Diagnosis and Prognosis of Gastrointestinal Diseases section Interesting images.

We appreciate the time and effort that you dedicated to provide feedback on our manuscript and we are grateful for the insightful comments on and valuable improvements to our paper and for considering interesting case of Gurvits syndrome with good endoscopic images.

We have incorporated your highly valuable suggestions in the manuscript, highlighted with track changes:

  1. The structure is difficult to read because there are no breaks throughout the text.

Response: The text was rearranged in a more readable manner.

  1. It is a type of article whose importance corresponds to the image. There is a lot of text that should be removed and summarized for easier reading .

Response: The text was adapted to the images, but also includes relevant details regarding the  background of this particular and rare type of pathology highlighting causes, stages and management. The patient has a complex pathologic status that gave rise to apparition of acute esophageal necrosis and there are more factors that are involved and found mentioned also  in the literature.

  1. I suggest providing a CT image

Response: According to valuable suggestion we added CT scan images relevant to patient particular condition

Sincerely yours, 

Sandica Bucurica

Round 2

Reviewer 2 Report

the authors have made a good correction adding the suggested changes

Author Response

Dear Reviewer,

We appreciate the time and effort that you dedicated to providing feedback on our manuscript and are grateful for the insightful comments on and valuable improvements to our paper.

Sincerely yours, 

Sandica Bucurica